# Optimisation of an Integrated System: Combined Heat and Power Plant with CO₂ Capture and Solar Thermal Energy

Agustín Moisés Alcaraz Calderón [1,2], Oscar Alfredo Jaramillo Salgado [2], Nicolas Velazquez Limón [3], Miguel Robles Perez [2], Jorge Ovidio Aguilar Aguilar [4], Maria Ortencia González Díaz [5] and Abigail González Díaz [1,*]

1   Instituto Nacional de Electricidad y Energías Limpias (INEEL), Cuernavaca 62490, Morelos, Mexico
2   Instituto de Energías Renovables, Universidad Nacional Autónoma de México, Temixco 62580, Morelos, Mexico
3   Universidad Autónoma de Baja California, Mexicali 21280, Baja California, Mexico
4   Universidad Autónoma del Estado de Quintana Roo, Chetumal 77039, Quintana Roo, Mexico
5   CONACYT–Centro de Investigación Científica de Yucatán, A.C., Mérida 97200, Yucatán, Mexico
*   Correspondence: abigail.gonzalez@ineel.mx

**Abstract:** This paper aims to evaluate different design configurations of a combined heat and power (CHP) plant with post-combustion CO₂ capture. Three cases are involved in this study: case 1 consists of three trains and each train has a configuration of one gas turbine with a heat recovery steam generator (HRSG); case 2 consists of three trains and one steam turbine; and case 3 consists of only two trains. The third case presented the highest CHP efficiency of 72.86% with 511.8 MW net power generation. After selecting the optimum configuration, a parabolic-trough collector (PTC) was incorporated to generate additional saturated steam at 3.5 bar for the capture plant, adding greater flexibility to the CHP because more steam was available. In addition, the efficiency of the cycle increased from 72.86% to 80.18%. Although case 2 presented lower efficiency than case 3, it has a steam turbine which brings the possibility of increasing the amount of electricity instead of steam production. When the PTC was incorporated in case 2, the power generated in the steam turbine increased from 23.22 MW to 52.6 MW, and the net efficiency of the cycle from 65.4% to 68.21%.

**Keywords:** combined heat and power; CO₂ capture; solar energy; parabolic-trough collector

## 1. Introduction

CO₂ emissions have been increasing greatly, with a high potential to produce catastrophic climate change. A new goal was established in the Paris Agreement related to limiting the temperature increase to 2 °C [1] Mexico has committed to mitigating its greenhouse gas emissions, e.g., those contained in the Kyoto Protocol and the United Nations Framework Convention on Climate Change. To fulfil the goal set by the Climate Change Act, Mexico has promised to mitigate "its greenhouse emissions by 50% below 2000 levels by 2050" [2].

The Mexican Ministry of Energy launched the new 2015–2029 Electricity Sector Foresight which included combined heat and power (CHP) for the first time, with an expected share of electricity generation of 6.8% by 2029 [3]. In 2019, the Mexican Federal Commission of Electricity (CFE) announced a plan to implement six cogeneration plants, the total capacity of which was 4392 MW and 4797 tonne/h of steam, with no progress to date [4]. In 2021, the expected generation with CHP in the Mexican plan accounted for 2.6% (2309 MW) [5]. Finally, in 2022, the share of CHP technology is expected to be around 3.39% from 2026–2036 [6].

Although CHP systems consist of a technology with low CO₂ emissions, in most cases they use natural gas combined with biomass. Using natural gas, the carbon emissions of CHP are around 250 kgCO₂/MW. Post-combustion carbon capture and storage (CCS) is an

alternative for decarbonising the electricity sector [7]. This technology would enable fossil fuel power plants to generate clean electricity from fossil fuels with low emissions [8]. A recent study showed that the Swedish CHP plants with CCS alone have the potential of reaching the goal of 11 Mton negative emissions [9]. This is because of the high thermal efficiency of the CHP.

In a CHP plant, because the demand for electricity and thermal energy are important, the plant must be flexible enough and ready to supply both when they are demanded. If a $CO_2$ capture unit is incorporated, $CO_2$ emissions are reduced but steam production becomes even more crucial because the post-combustion capture plant requires steam to regenerate the solvent. One option for CHP with CCS is supplementary firing, which makes it more flexible and gives greater control over the electricity and thermal energy separately [10]. Additionally, supplementary firing is a technology which is widely used to compensate for electrical demand, when it is reduced due to the intermittency of renewable energy [11,12]. Small-scale carbon capture incorporated into micro-combined heat and power co-generation systems has the potential to reduce carbon emissions [13]. However, both sequential supplementary firing and the microturbine CCS system are penalised by the extraction of steam to regenerate the solvent.

Several hybrid systems have been proposed by several authors. Bioenergy with CCS (BECCS) is recognised as a negative emissions technology which could be applied in a CHP. However, the energy penalty incurred in power plants makes BECCS unattractive [14]. A case study for Stockholm was evaluated by [15], to analyse the insights into barriers and policy implications in relation to successful BECCS implementation.

Ref. [16], analysed a CHP with carbon capture and utilization. The $CO_2$ is used to generate methane via the methanation process, which used green hydrogen produced through electrolysis and electricity obtained from solar energy. Another work by the same author [17], was a CHP fuelled with blends of natural gas and hydrogen. However, large investment is needed in the first alternative; in the second, the percentage of hydrogen is limited by the combustor, which is a barrier for existing plants.

A recent study published by [18], proposed a CHP with CCS and compressed $CO_2$ energy storage. According to the results, this system improved power generation and efficiency, but thermal energy is obtained from the power plant. Another alternative that can be incorporated in a CHP is geothermal-energy-assisted CCS. Geothermal energy has the advantage of non-intermittence compared with solar and wind energy [19]. However, geothermal energy is available only in specific regions.

A hybrid system, in which solar thermal technology is incorporated into a CHP with CCS, is an alternative for generating steam for the capture plant without compromising the entire steam generation for the process and the efficiency of the system [20]. This integrated system could play an important role in the transition to a sustainable energy economy and for abating $CO_2$ emissions from existing conventional power, as well as for industrial sectors [21,22]. It is clear that additional investment will be required, but the cost of solar energy technology has been reduced significantly [23]. A solar thermal plant collects sunlight with the help of concentrators. The sunlight can directly heat water and turn it into steam to regenerate the amine solvent. One disadvantage is the fact that solar energy depends on bright sunshine, can be solved with thermal energy storage. [24], evaluated the applicability of solar and wind energy sources in a CHP system for small and distributed communities. One of their conclusions was that the integrated system is a sustainable, and socio-economically and environmentally feasible energy management solution.

The annual average direct solar irradiance in Mexico is shown in Figure 1. In Baja California and part of the central area of the country, it is between 5.66 and 6.16 kWh/m$^2$ day. Given the high solar radiation in Mexico, this technology would be a good option for incorporating into a CHP.

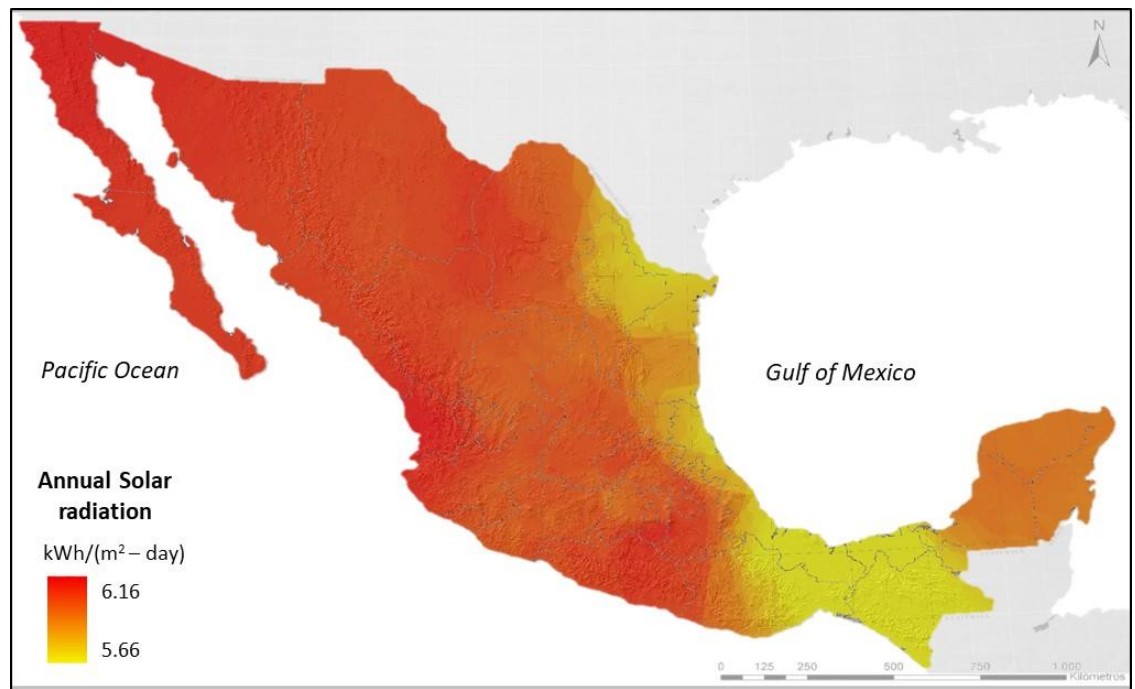

**Figure 1.** Annual average direct normal irradiance in Mexico.

Some work related to hybrid systems has been developed. [25], proposed a combined system: solar energy and post-combustion capture technology without energy storage. The solar energy was provided to the system only as a small proportion of the energy required for solvent regeneration, and particularly only during the day. [26], developed a techno-economic study of an integrated system: post-combustion $CO_2$ capture integrated with a coal power plant. The thermal energy for regenerating the solvent was generated using three alternatives: PTC, a Stirling dish collector (SDC), and a solar power tower (SPT) to generate steam for PCC. It was concluded that the PTC and SPT are technically viable for providing thermal energy for the PCC. In addition, low-temperature solar thermal systems could be better integrated with solvent extraction equipment. A techno-economic feasibility study of a $CO_2$ capture plant assisted by solar thermal energy with Fresnel technology was developed by [27], for a 300 MW coal power plant which would be located in New South Wales, Australia. However, this study considered thermal storage with sensitive solids and monoethanolamine (MEA). Other studies related to CCS and solar energy integrated with a coal power plant were published by [28].

Ref. [29], demonstrated that optimising design strategies in cogeneration reduces the cost of $CO_2$ [30], optimised a CHP by utilising low pressure steam and the waste heat of the plant. [31], continued with the previous work to demonstrate how CHP configurations can be utilised to reduce the cost of production not only by using electricity and steam, but also $CO_2$ for enhanced oil recovery (EOR). [32], assessed the potential and constraints of incorporating PCC into a cogeneration plant.

Ref. [33], evaluated the integration of solar and CCS into a micro cogeneration plant, the main equipment of which was the Capstone microturbine with a capacity of 200 kW and the oil heat recovery. [34], published a comparative techno-economic analysis of: 1. a CHP assisted with steam production using solar energy; and 2. a conventional CHP with PCC. The results showed that option 2, cogeneration with PCC, had a lower levelised cost of steam (LCS) than the one produced from solar technology. None of these works considered optimising the configuration of the cogeneration process to generate steam before incorporating CCS and solar energy, which is the intention of this work.

Refs. [35–37], published studies of a natural gas combined cycle power plants (NGCC); CCS, solar thermal, hydrogen, and hot water in a dual-pressure organic ranking cycle were integrated to optimise the heat and improve the efficiency.

*Novelty*

This paper evaluates the potential of incorporating MEA-based $CO_2$ capture in a CHP, assisted by a parabolic-trough collector (PTC) with thermal energy storage (TES) using a thermal fluid.

Firstly, the article includes a quantitative analysis of the impact of the configurations of the CHP with PCC. Three case studies are evaluated to define the one with the highest efficiency, at a constant steam demand of 700 tonne/h and leaving electricity generation free.

Secondly, after selecting the case with the highest efficiency, this work proposes the incorporation of a PTC to increase the power output, efficiency, and operating flexibility to supply steam and electricity. The PTC generates steam for regenerating the amine solvent of the PCC plant, as presented in Figure 2. Thus, the efficiency of the CHP plant is mostly penalised as a result of compressing the $CO_2$.

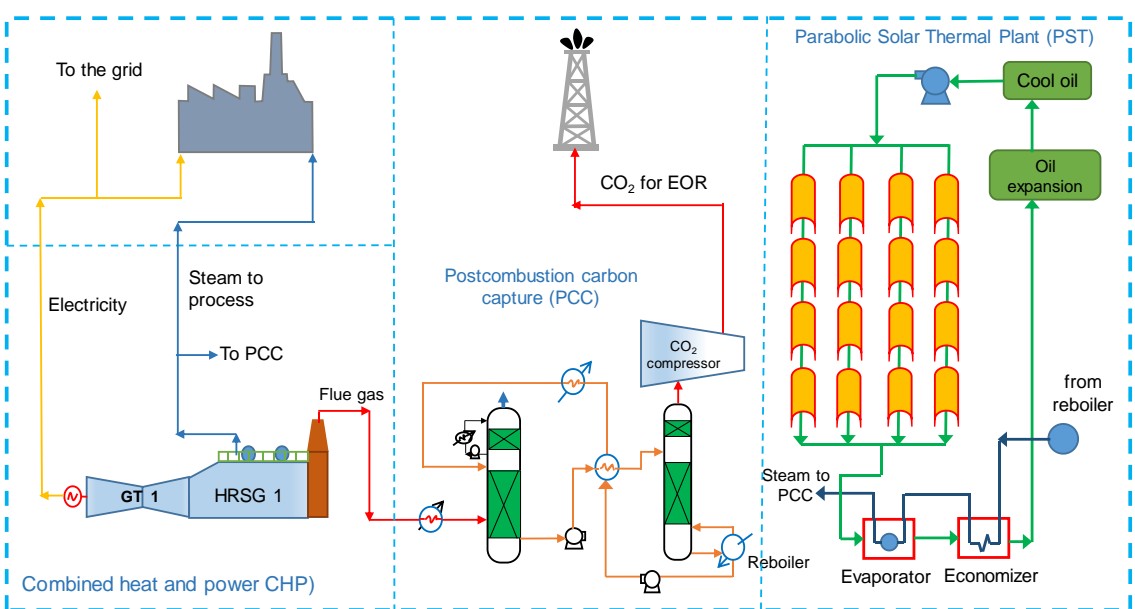

**Figure 2.** Schematic process of a natural gas combined heat and power plant configuration with a gas turbine, double pressure HRSG, a post-combustion MEA-based $CO_2$ capture, assisted by solar energy and thermal energy storage using a thermal fluid.

Although an economic assessment is identified as of great importance and should be investigated in future research, a detailed cost analysis of the integrated system goes beyond the scope of the current study.

## 2. Combined Heat and Power with CCUS

### 2.1. Combined Heat and Power Plant

To evaluate and define the optimum configuration of CHP with MEA-based $CO_2$ capture that represents the highest efficiency, three configurations were evaluated, keeping the production of steam for the process at 700 tonne/h and leaving electricity generation free. Considering that, excess electricity could be sold to the grid. The three alternatives are:

1.  Three trains of CHP, with the configuration of each train consisting of a GE 7F04 gas turbine (GT) connected to a heat recovery steam generator (HRSG), as shown in Figure 3. The flue gas exiting the gas turbine enters the HRSG, where intermediate pressure steam is generated for the petrochemical process and for the PCC. Addi-

tional steam at low pressure is produced in the HRSG to increase the CHP plant's power generation.

2.　Three trains of CHP, with the configuration of each train consisting of a GE 7F05 gas turbine with a HRSG as shown in Figure 4. The flue gas leaving the gas turbine enters the HRSG, where intermediate pressure steam is generated, but in this case, a portion of the steam feeds into a steam turbine and the remainder goes to the petrochemical process and the capture plant. Additional low pressure steam is produced in the HRSG to increase the CHP plant's power generation.

3.　Two trains of CHP, with each train consisting of a GE H01 gas turbine with a HRSG, as shown in Figure 5. The GTs have higher capacity than cases 1 and 2. The flue gas leaving the GT enters the HRSG, where intermediate pressure steam is produced for the petrochemical process and for the PCC. In addition, steam at low pressure is produced in the HRSG to increase the CHP plant's power generation.

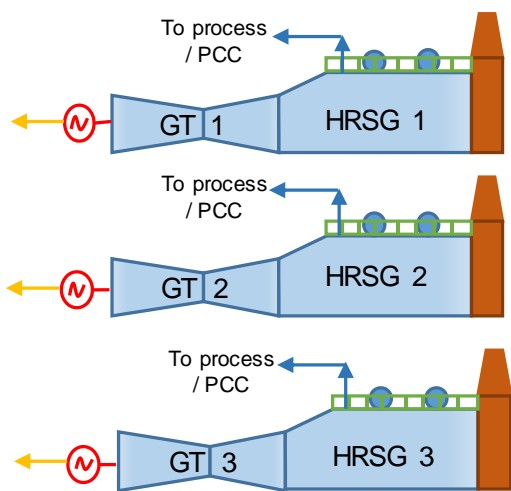

**Figure 3.** Case 1. Three trains of combined heat and power, with the configuration of each train consisting of one gas turbine, GE 7F04, with a HRSG.

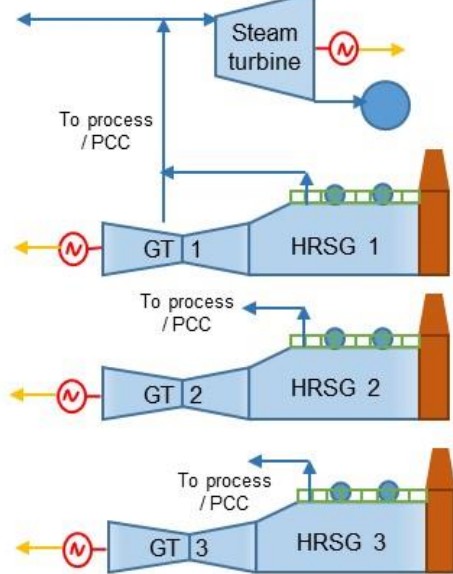

**Figure 4.** Case 2. Three trains of combined heat and power, with the configuration of two trains consisting of one gas turbine GT with a HRSG; and one train consisting of one gas turbine, GT 7F05, with a HRSG and one steam turbine.

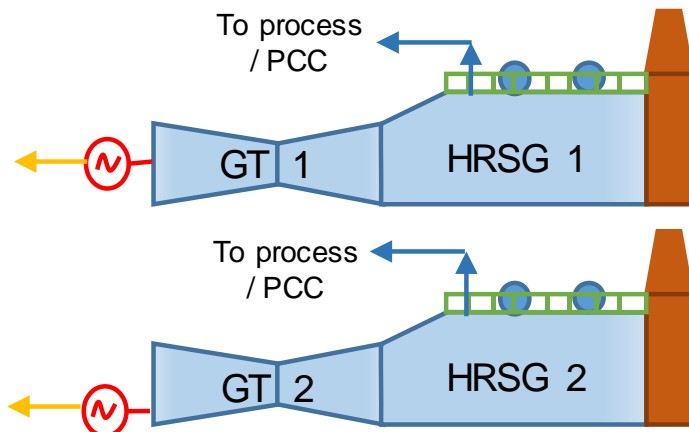

**Figure 5.** Case 3. Two trains of combined heat and power, with the configuration of each train consisting of one gas turbine, GE 7H.01, with a HRSG.

The gas price reduction, higher efficiency, lower capital costs, and minimal SOx emissions, has led to a significant increase in the number of NGCC and CHP plants in Mexico, in the last years [4]. For that reason, GT integrated in a HRSG is selected in this study.

The mass and energy balances with CCS were estimated as follows:

1. The gas turbines were simulated using GE GTP software from General Electric.
2. The amount of steam generated was estimated based on the heat and mass balances modelled in Excel using the free steam tables based on IAPWS-IF97 (Industrial Formulation).

The first equation is the energy balance between the gas (hot) and steam (cold) streams. Both heat loss by radiation and convection were considered in each section of the HRSG, which is presented by the simplified Equations (1) and (2).

$$Q_{in} = \dot{m}_s(h_{sout} - h_{sin}) \tag{1}$$

$$Q_{in} = \dot{m}_g(h_{gout} - h_{gin}) \tag{2}$$

where $Q_{in}$ is the heat absorbed by the steam in kW, $h_s$ and $h_g$ are the steam and gas side enthalpies, respectively, in (kJ /kg), and $\dot{m}_s$ and $\dot{m}_g$ are the steam and gas mass flow in (kg/s).

The cogeneration efficiency $\eta_{CHP}$ in (%) was estimated using Equation (3).

$$\eta_{CHP} = \frac{Qt + W}{\dot{m}_g LHV} \tag{3}$$

where $Qt$ is the total heat (steam), $W$ is the net electric power in (MW), $\dot{m}_g$ is the mass flow rate of the natural gas in (kg/s), and *LHV* is the natural gas low heat value (MJ/kg).

Ambient conditions, the natural gas composition, the *LHV*, as well as the pressure, temperature, and the amount of the steam required in the process considered in this work are given in Table 1.

**Table 1.** Ambient conditions and fuel composition.

| Ambient Condition | | |
|---|---|---|
| Pressure | bar | 0.98 |
| Temperature | °C | 33 |
| Relative humidity | % | 40 |
| Steam required in the process | tonne/h | 700 |
| Steam temperature | bar | 350 |
| Steam pressure | °C | 19.6 |
| Natural gas composition | | |
| Methane | % mol | 94.59 |
| Ethane | % mol | 3.89 |
| Propane | % mol | 0.205 |
| n-buthane | % mol | 0.026 |
| n-pentane | % mol | 0.016 |
| n-hexane | % mol | 0.051 |
| Nitrogen | % mol | 0.148 |
| Carbon dioxide | % mol | 1.074 |
| HLV | kJ/kg | 48,318 |
| Molar weight | kmol/kg | 17.01 |

### 2.2. Post-Combustion MEA-Based $CO_2$ Capture Plant

The three case studies were integrated with a post-combustion MEA-based $CO_2$ capture at 30 wt%, as shown in Figure 1.

The $CO_2$ capture plant was simulated using Aspen Plus® V11 from Aspentech company, the rate-based approach was used. The rate-based model provides excellent predictions for the overall performance of the capture plant, e.g. the lean loading, energy consumption in the reboiler, that cannot be predicted using the equilibrium-stage model. In addition, the rate-based model is a very useful optimization tool to study sensitivities of various $CO_2$ capture process variables, as described in [4].

The optimum values were obtained for the most important parameters, e.g., solvent lean loading, solvent rich loading, and thermal energy to reach 90% $CO_2$ capture rate using a height of the absorber packing of 21 m [4]. A summary of the steps to optimise the design of the PCC system based on [38,39], is as follows:

- Variation of the MEA lean solvent loading solution to define the minimum energy required in the reboiler for a specific $CO_2$ concentration of the flue gas.
- While studying the impact of different lean loading on the capture system, the pressure in the reboiler is varied in order to adjust the lean loading. The temperature is maintained constant at 120 °C.
- The solvent circulation rate in the absorber is changed to obtain 90% $CO_2$ capture. Flue gas at the inlet of the absorber is 44 °C and the pressure is 1.13 bar; and $CO_2$ leaves the stripper condenser at 40 °C.

### 2.3. Performance of the Three Case Studies

The performance of the CHP plant with the $CO_2$ capture plant is presented in Table 2. The configurations and operating parameters for the three cases were taken from [4]. The integration of the CHP and the PCC consisted of the steam which was extracted from the pipe between the HSRG and the process at 20 bar, which was reduced to 3 bar to regenerate the MEA solvent.

**Table 2.** Summary of key parameters of the three cases of CHP with post-combustion carbon capture (90% capture).

| Concept | Case 1 | Case 2 | Case 3 |
|---|---|---|---|
| Net power output (MW) | 493.7 | 635.85 | 511.8 |
| Gas turbine power (MW) | 493.7 | 612.6 | 511.8 |
| Steam turbine power (MW) | 0 | 23.22 | 0 |
| Natural gas consumption (MW) | 1362.6 | 1849.3 | 1344.4 |
| Steam mass flow to the process (tonne/h) (20 bar and 350 °C) | 700 | 700 | 700 |
| Flue gas composition (% vol.) | | | |
| Ar | 0.87 | 0.87 | 0.87 |
| $N_2$ | 73.51 | 73.55 | 73.27 |
| $O_2$ | 11.85 | 11.96 | 11.16 |
| $CO_2$ | 4.05 | 4.00 | 4.37 |
| $H_2O$ | 9.72 | 9.62 | 10.33 |
| Flue gas flow rate (tonne/h) | 4381.9 | 5391.9 | 4000.7 |
| Post-combustion (MW) | 9.78 | 0 | 77.3 |
| Steam required for the capture plant (tonne/h) (Saturated at 3.5 bar) | 412 | 500.87 | 406.7 |
| CHP efficiency + PCC + $CO_2$ compression (%) | 70.53% | 65.04% | 72.86% |

Intermedia pressure steam (20 bar and 350 °C) was generated for the process as well as for the PCC. The pressure of the intermedia pressure steam was reduced from 20 bar to 3.5 bar and was attempered to obtain saturated steam. Case 3 presented the highest CHP efficiency of 72.86%, generating 511.8 MW of net power output. This was followed by case 2, with a 65.04% efficiency and 635.85 MW, and by case 1, at 70.53% generating 493.7 MW. The efficiency increased because of the flue gas flow rate as well as its $CO_2$ concentration. As a result, less $CO_2$ is captured and then less steam is required to regenerate the solvent; 406.7 tonne/h, 500.87 tonne/h, and 412 tonne/h for case 3, case 2, and case 1, respectively. Although efficiency is important, it is very important to consider the availability for selling power to grid. Case 3 was selected to incorporate a solar thermal plant to generate steam for the capture plant and then to improve the efficiency of the CHP. In addition, leaving the steam generated in the CHP only for the petrochemical plant will bring more flexibility to the whole system.

The capture plant optimisation of the three case studies is presented in Figures 6–8. As mentioned in [4], the higher the $CO_2$ concentration in the flue gas, the lower the energy in the reboiler. This is because the higher rich loading attained with higher $CO_2$ concentration conducts to an increment in solvent capacity, and the reboiler duty reduces. As presented in Figure 6, the optimum energy in the reboiler for case 1 is 3.78 MJ/kg$CO_2$, heat is 258 MW, and the optimum pressure in the stripper is 1.84 bar at 120 °C. The flue gas $CO_2$ concentration is 4.05 mol%, at a 90% $CO_2$ removal rate.

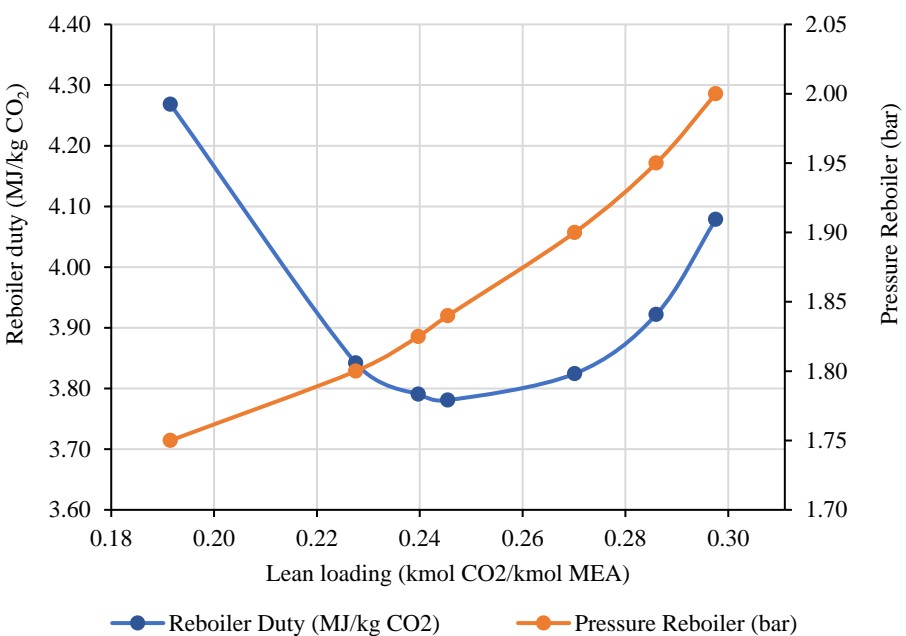

**Figure 6.** Optimisation of the energy in the reboiler of the capture plant as a function of solvent lean loading for case 1. Stripper temperature is 120 °C and the flue gas $CO_2$ concentration is 4.05 mol%, at 90% $CO_2$ removal rate.

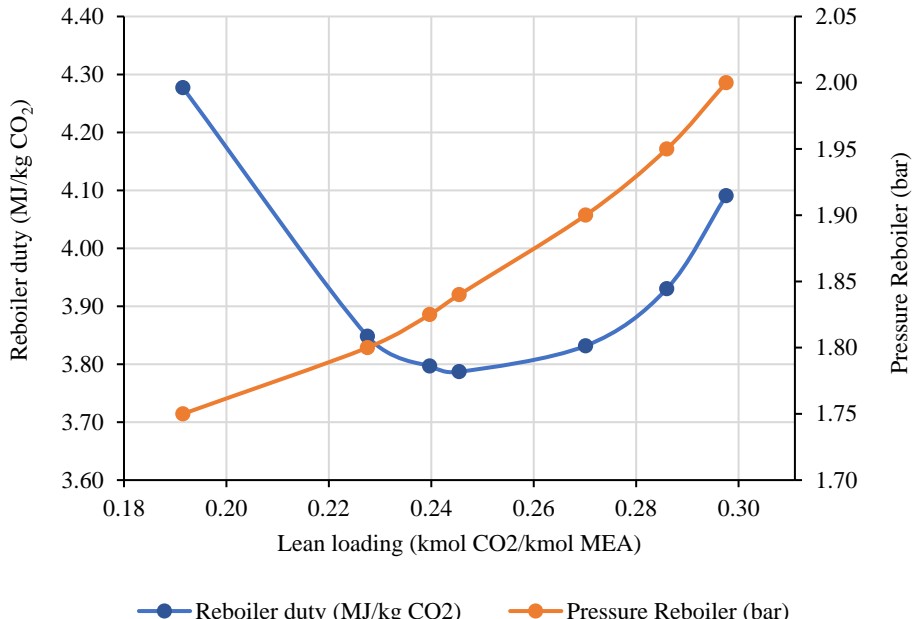

**Figure 7.** Optimisation of the energy in the reboiler of the capture plant as a function of solvent lean loading for case 2. Stripper temperature is 120 °C and the flue gas $CO_2$ concentration is 4.0 mol%, at 90% $CO_2$ removal rate.

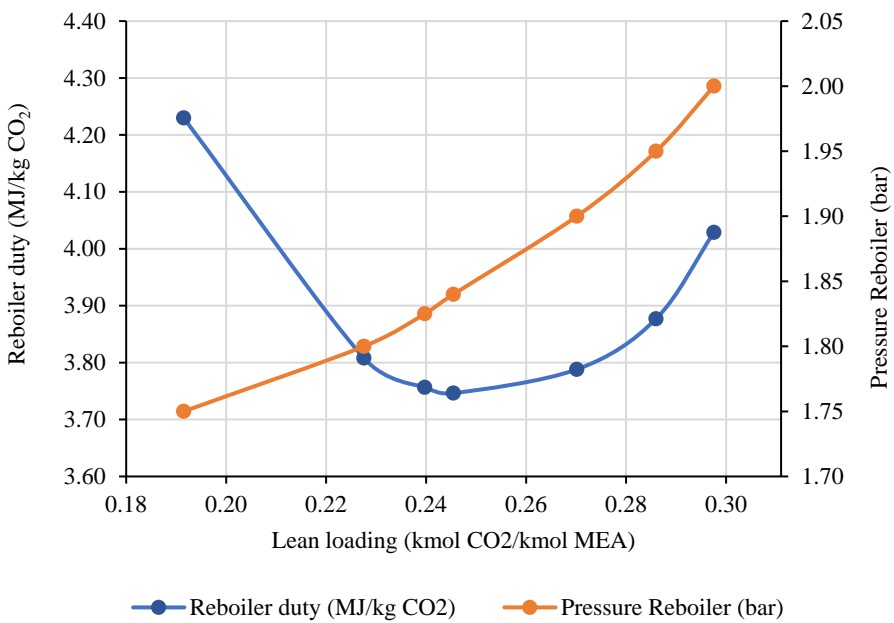

**Figure 8.** Optimisation of the energy in the reboiler of the capture plant as a function of solvent lean loading for case 3. Stripper temperature is 120 °C and the flue gas $CO_2$ concentration is 4.37 mol%, at 90% $CO_2$ removal rate.

As presented in Figure 7, the optimum energy in the reboiler for case 2 is 3.79 MJ/kg$CO_2$, heat is 314 MW, and the optimum pressure in the stripper is 1.84 bar at 120 °C. The flue gas $CO_2$ concentration is 4.0 mol%, at a 90% $CO_2$ removal rate.

As presented in Figure 8, the optimum energy in the reboiler for case 3 is 3.75 MJ/kg$CO_2$, heat is 252 MW, and the optimum pressure in the stripper is 1.84 bar at 120 °C. The flue gas $CO_2$ concentration is 4.37 mol%, at a 90% $CO_2$ removal rate.

In case 3, the flue gas had the highest $CO_2$ concentration and represented the lowest energy requirement in the reboiler of 3.75 MJ/kg$CO_2$ and thermal heat of 252 MW, as shown in Figure 8. In comparison, these figures were 3.78 MJ/kg$CO_2$ and 258 MW in case 1, and 3.79 MJ/kg$CO_2$ and 314 MW in case 2, with $CO_2$ concentrations of 4.05 vol% and 4 vol%, respectively. The absorber column packing height considered in all three cases was 21 m.

The stripper packing dimensions and the resulting absorber and stripper diameters for all cases are presented in Table 3.

**Table 3.** Absorber and stripper packing dimensions.

| Parameter | Unit | Case 1 | Case 2 | Case 3 |
|---|---|---|---|---|
| Number of trains | | 2 | 2 | 2 |
| Absorber packing heigh | meter | 21 | 21 | 21 |
| Absorber diameter | m | 15.83 | 17.54 | 15.27 |
| Stripper packing heigh | m | 13 | 13 | 13 |
| Stripper diameter | m | 7.15 | 7.16 | 7.12 |

## 3. Combined Heat and Power with $CO_2$ Capture and Solar Energy for Solvent Regeneration

Case 3, a CHP plant with $CO_2$ capture, shown in Figure 5, was integrated with solar thermal power, which generates saturated steam at 3 bar for solvent regeneration. Two options were considered for solvent regeneration using solar thermal power:

Hybrid alternative: Solar panels are used to produce steam for solvent regeneration during the day and steam extracted from the power plant during the night.

Solvent storage at 100% during the night and regeneration during the day: In this alternative, solar panels with double the capacity are required, as well as a stripper with double capacity in order to regenerate the storage solvent plus the solvent used during the day.

The System Advisor Model (SAM) software was selected to simulate the parabolic-trough solar plant. It is used to develop a techno-economic analysis of photovoltaic, battery storage, concentrating solar, parabolic trough, power tower, and other systems [40].

The information required to simulate the solar concentrator for steam generation is presented in Table 4.

**Table 4.** Design parameters for the simulation.

| Location and Resource | Values | Reference | Solar Field–Continue | Values | Reference |
|---|---|---|---|---|---|
| Latitude and longitude. | 25.61 DD and −99.98 DD | NREL, NSRDB | Header design maximum flow velocity | 3 m/s | NREL, SAM |
| Elevation | 324 m, | NREL, NSRDB | Collector tilt | 0° | Assume |
| Global horizontal irradiation, direct normal (beam) irradiation and diffuse horizontal Irradiation: | 5.36 kWh/m$^2$/day, 5.61 kWh/m$^2$/day and 1.74 kWh/m$^2$/day | NREL, NSRDB | Collector azimuth | 0° | NREL, SAM |
| Average temperature: | 22.1 °C | NREL, NSRDB | Stow angle | 170° | NREL, SAM |
| Average wind speed: | 2.5 m/s | NREL, NSRDB | Deploy angle | 10° | NREL, SAM |
| **System Design** | | | Water usage per wash | 0.7 L/m$^2$ aperture | NREL, SAM |
| Design point direct normal irradiance | 950 W/m$^2$ | NREL, SAM | Washes per year | 12 | NREL, SAM |
| Target solar multiple | 2.5 | NREL, SAM | Hot piping thermal inertia | 0.2 kWht/K-MW-t | NREL, SAM |
| Target receiver thermal power | 743.05 MWth | Assume | Cold piping thermal inertia | 0.2 kWht/K-MW-t | NREL, SAM |
| Loop inlet heat transfer fluid temperature | 150 | Assume | Field loop piping thermal inertia | 4.5 Wht/K-m | NREL, SAM |
| Loop outlet heat transfer fluid temperature | 350 | Assume | Non solar field land area multiplier | 1.1 | NREL, SAM |
| Heat sink power | 296.7 MWt | Assume | **Solar collectors assembly** | **Values** | **Reference** |
| Pumping power for Heat Transfer Fluid through heat sink | 0.55 kW/kg/s | NREL, SAM | Collector type selection. | FLABEG Ultimate Trough RP6 (with 89-mm OD receiver for oil HTF) | Assume |
| Hours of storage at design point | 6 | NREL, SAM | **Receiver selection or heat collection element** | | |
| **Solar field** | | | Receiver selection | Schott PTR70 | Assume |
| Row spacing | 15 m | NREL, SAM | **Thermal storage** | | |
| Header Pipe roughness | 4.57 × 10$^{-5}$ m | NREL, SAM | Tank height | 15 m | NREL, SAM |
| HTP pump efficiency | 85% | NREL, SAM | Tank fluid minimum height | 0.5 m | NREL, SAM |
| Piping thermal loss coefficient | 0.45 W/m$^2$ K | NREL, SAM | Parallel tank pairs | 1 | NREL, SAM |
| Wind stock speed | 25 m/s | NREL, SAM | Water loss coefficient | 0.3 Wt/m$^2$-K | NREL, SAM |
| Receiver startup delay time | 0.2 h | NREL, SAM | Initial hot Heat Transfer Fluid percent | 30% | NREL, SAM |

**Table 4.** *Cont.*

| Location and Resource | Values | Reference | Solar Field–Continue | Values | Reference |
|---|---|---|---|---|---|
| Receiver startup delay energy fraction | 25% | NREL, SAM | Cold tank heater temperature set point | 60 °C | NREL, SAM |
| Collector startup energy | 0.021 kWhe/SCA | NREL, SAM | Cold tank heater capacity | 0.5 MWe | NREL, SAM |
| Tracking power per Solar Collector Assembly | 125 W/SCA | NREL, SAM | Hot tank heater temperature set point | 110 °C | NREL, SAM |
| Field heat transfer Fluid | Therminol VP-1 | Assume | Hot tank heater capacity | 1 MWe | NREL, SAM |
| Freeze protection temperature | 12 °C | NREL, SAM | Tank heater efficiency | 0.99 | NREL, SAM |
| Minimum single loop flow rate | 1 kg/s | NREL, SAM | **System control** | | |
| Maximum single loop flow rate | 12 kg/s | NREL, SAM | Fraction of rated gross power consumed all times | 0.0055 MWe/ MWtcap | NREL, SAM |
| Header design minimum flow velocity | 2 m/s | NREL, SAM | Balance of plant parasitic | 0 MWe/MWtcap | NREL, SAM |

## 4. Plant Configuration/Specification

To evaluate plant performance (e.g., total thermal energy, efficiency, capacity factor, and total thermal energy taken by the concentrated solar power plant), the design parameters considered as inputs in the SAM software simulation are based on [41] and are presented in Table 4.

The plant was simulated from 0 h to 8760 h, which represents a whole year. For the simulation of the concentrated solar energy (CSE) plant, at the site with high direct normal irradiance (DNI) and the typical metrological year (TMY), information from the NREL database was used to analyse the performance of the parabolic-trough collector solar plant (PTCSP).

Climate data includes hourly DNI, wind speed, atmospheric pressure, ambient temperature, solar azimuth angle, and sun angle for the whole year. Mexico receives a medium solar DNI range, which varies from 5.66 kW h/m$^2$/day to 6.16 kW h/m$^2$/day [42].

The annual thermal energy production in kW as a function of time is presented in Figure 9. As can be noted, the highest thermal energy generation is generated between 9 am and 6 pm, considering the incorporation of 6 h of energy storage.

PTCSP technology is economically feasible if the DNI is greater than 5.5 kW h/m$^2$/day. Cadereyta, in the state of Nuevo León, Mexico, was selected as the location of the CHP plant with carbon capture due to its favourable annual average DNI solar conditions, with a maximum in August of 6.6 kW h/m$^2$/day and a minimum in December of 3.7 kW h/m$^2$/day, as presented Figure 10. In addition, there is a refinery located in the city. Refineries are used to incorporate a CHP plant. The direct irradiance heat map (W/m$^2$) for the whole year is shown in Figure 11. Table 5 presents the results of the systems, such as the annual energy, annual thermal freeze protection, capacity factor, annual electricity load and total field area (ha).

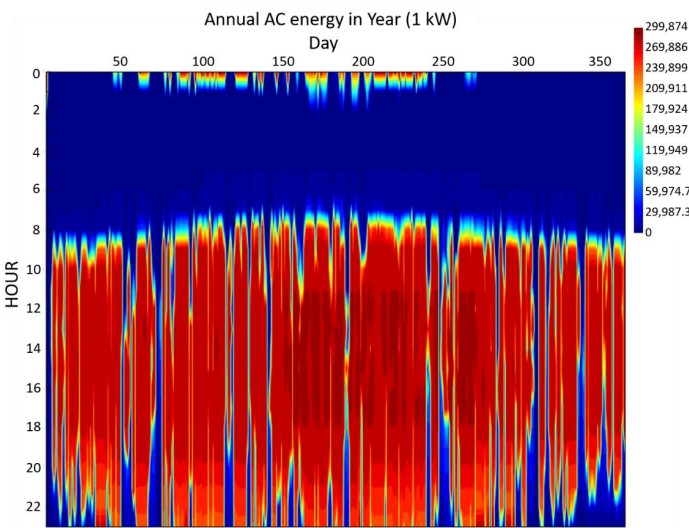

**Figure 9.** Annual energy production as function of time (kW).

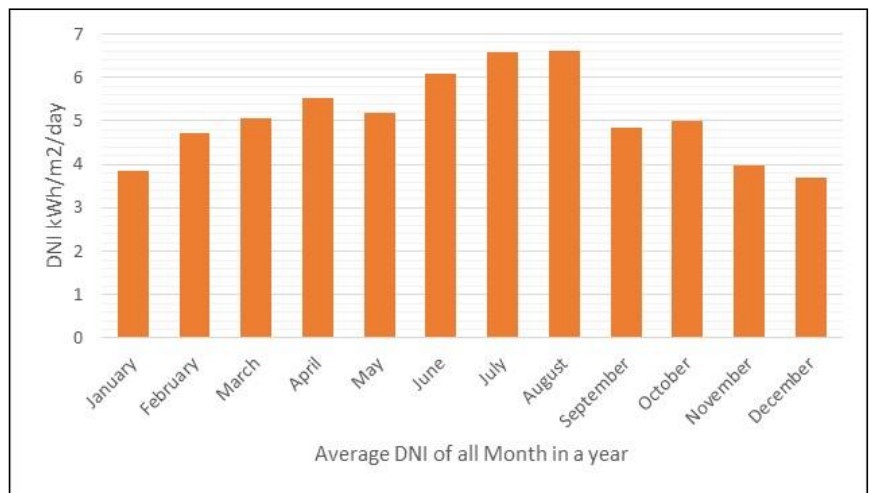

**Figure 10.** Monthly average DNI for each month of the year available at Cadereyta, Nuevo León, Mexico.

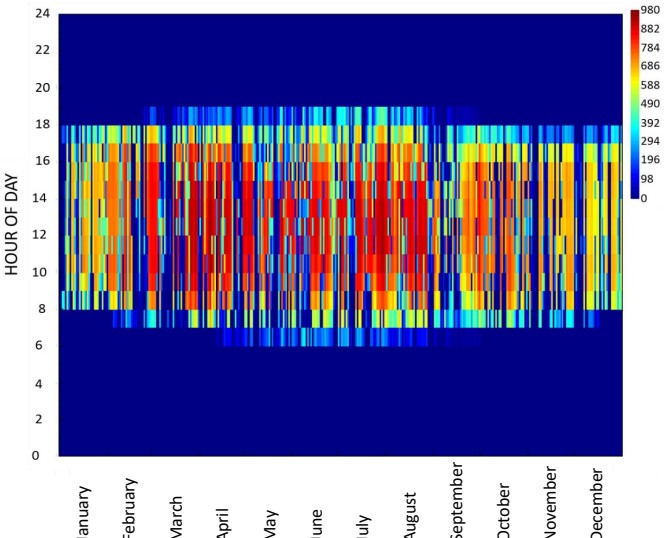

**Figure 11.** Heat map of annual direct irradiance in $W/m^2$ during 24 h of a day.

**Table 5.** Performance metrics.

| Metric | Value |
|---|---|
| Annual energy (year 1) | 910,011.53 MWh-t |
| Annual thermal freeze protection (year 1) | 0 kWh-t |
| Capacity factor | 41.1% |
| Annual electricity load (year 1) | 5,930,754 kWh-e |
| Total field area (ha) | 224.6 |

*System Performance*

The monthly thermal power incident of the PTC is presented in Figure 12. As can be seen, the maximum thermal power incident was in August, with 212,614 MWh, and the minimum was in November, with 156,505 MWh. In the same figure, system thermal heat is presented, with the maximum in August, with 119,140 MWh, and the minimum in December, with 62,872 MWh. When the thermal heat circulates from the cold tank to the hot tank, the minimum temperature of the cold header inlet reached a value of 149.0 °C. Similarly, the maximum temperature obtained at the hot header outlet was 249.87 °C.

In Mexico, the maximum solar radiation is around 6.16 kWh/m$^2$ for 10–12 h/day throughout the year. The maximum cycle efficiency obtained from the plant was 85%.

The temperature loss from the HTF fluid, Therminol VP-1, is 1 °C/h approximately, which is reduced over time. The maximum cycle thermal energy input was recorded as 119,140 MWt during the month of August. Similarly, the maximum field thermal energy incident was recorded as 212,614 MWt. Thermal energy generation depends on the field thermal energy incident, as shown in Figure 13. To generate thermal energy after sundown, the HTF fluid must be stored in the tank with a storage capacity of 6 h. The maximum total volume considered related to the TES HTF tank is 8783 m$^3$, while it is 8039.5 m$^3$ for the TES HTF hot tank, and 8039.5 m$^3$ for the TES HTF cold tank.

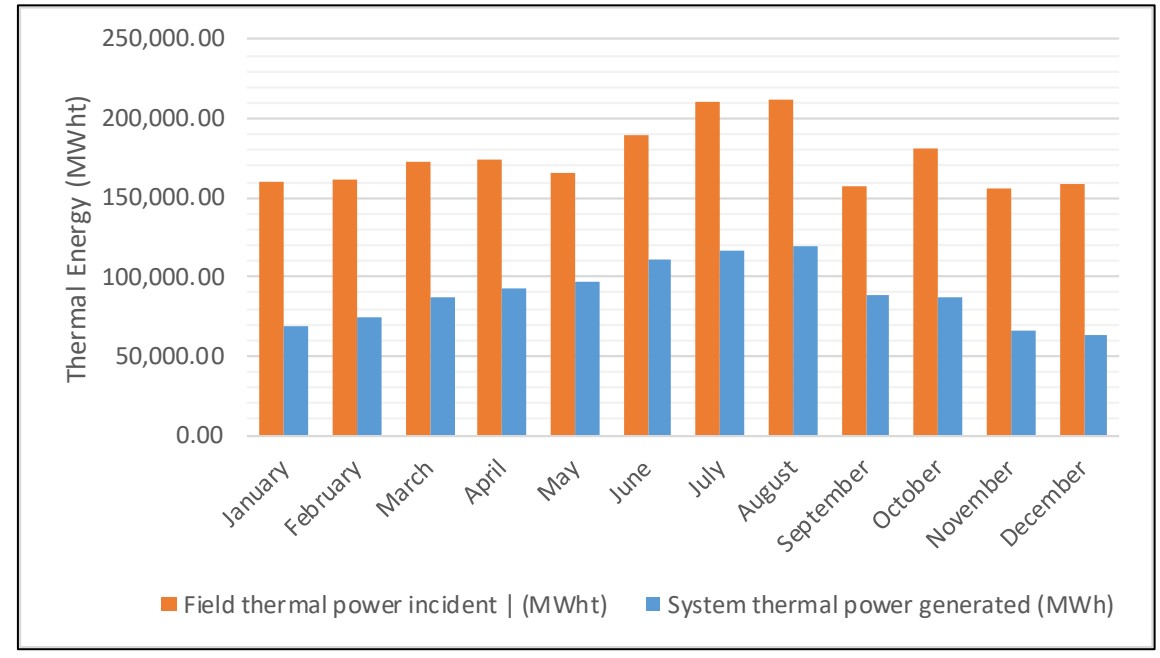

**Figure 12.** Monthly thermal power incident and power thermal generated from the parabolic trough concentrating solar thermal power plant (PTCSTPP).

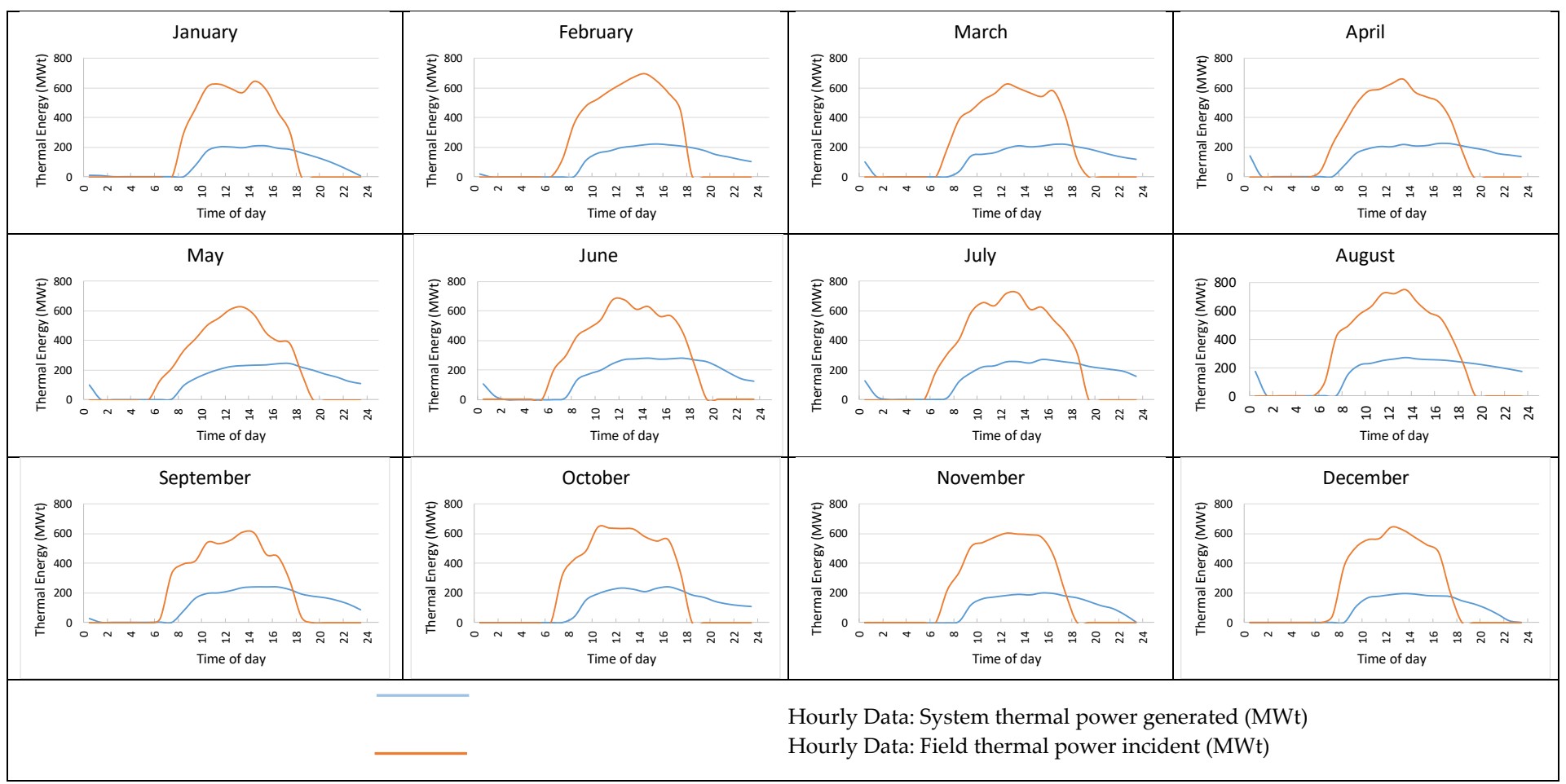

**Figure 13.** Hourly data for System Thermal Energy Generated and Field Thermal Energy Incident.

Without the PTC, as shown in Figure 14, the total steam generated from the CHP plant was 1052 tonne/h of steam at 350 °C and 20 bar. Of this, 700 tonne/h was for the process and the other 352 tonne/h was sent to the capture plant, but it first passes through a valve to reduce its pressure to 3.5 bar. The 352 tonne/h was attempered with 54.7 tonne/h of water in order to decrease the temperature.

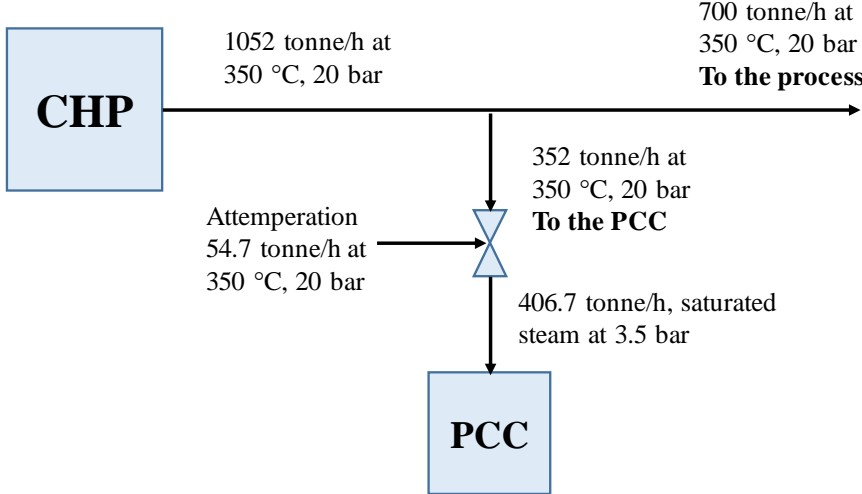

**Figure 14.** Steam mass balance of the CHP plant with $CO_2$ capture of case 3.

When the PTC was incorporated (case 3), as presented in Figure 15, instead of sending 352 tonne/h to PCC from the CHP plant, this was reduced to 201.3 tonne/h. Thus, 850 tonne/h was sent to the process. For this reason, the CHP efficiency with PPC increased from 72.86% to 80.18%, as presented in Table 6. In relation to $CO_2$ emissions, it is important to mention that in a conventional CHP plant (without PCC), the carbon intensity was 220 $kgCO_2/MW$. When PCC was incorporated, it was reduced to 27.84 $kgCO_2/MW$, and when the PTC is incorporated, it was further reduced from 27.84 $kgCO_2/MW$ to 25.29 $kgCO_2/MW$.

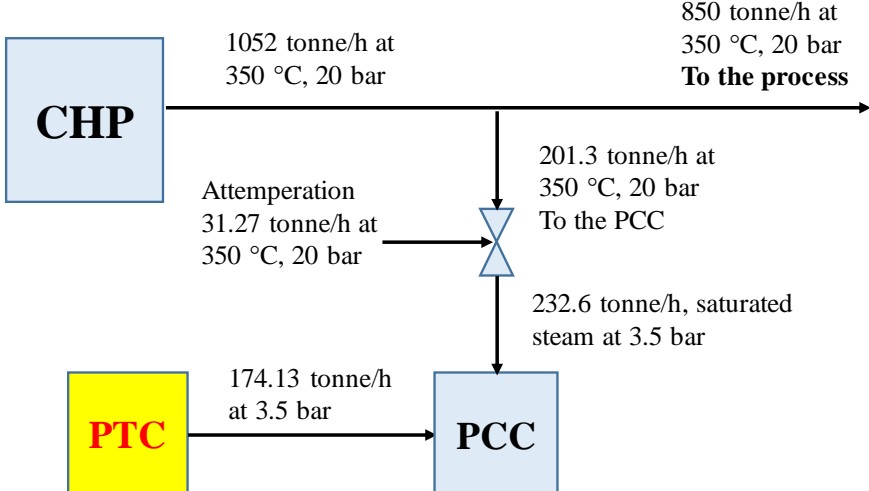

**Figure 15.** Steam mass balance of the CHP plant with $CO_2$ capture and the PTC of case 3.

**Table 6.** Case three with solar energy concentrator.

| Concept | CHP with PCC | CHP with PCC and PTC |
|---|---|---|
| Net power output to grid (MW) | 511.8 | 511.8 |
| Gas turbine power (MW) | 511.8 | 511.8 |
| Steam turbine power (MW) | 0 | 0 |
| Electric consumption compression (MW) | 20.1 | 20.1 |
| Net Power output to grid after compression (MW) | 491.7 | 491.7 |
| Natural gas consumption (MW) | 1344 | 1344 |
| Steam mass flow to the process (tonne/h) (20 bar and 350 °C) | 700 | 826 |
| Additional steam mass flow generated in the PTC (tonne/h) | | |
| Flue gas composition (% vol.) | | |
| Ar | 0.87 | 0.87 |
| $N_2$ | 73.27 | 73.27 |
| $O_2$ | 11.16 | 11.16 |
| $CO_2$ | 4.37 | 4.37 |
| $H_2O$ | 10.33 | 10.33 |
| Flue gas flow rate (tonne/h) | 4000.7 | 4000.7 |
| Post-combustion (MW) | 77.3 | 77.3 |
| Steam for the capture plant (tonne/h) (Saturated at 3.5 bar) | 406.7 | 406.7 |
| CHP efficiency + PCC + $CO_2$ compression (%) | 72.86% | 80.18% |
| Carbon intensity of the CHP (kg $CO_2$/MW) without PCC | 220 | 220 |
| CHP $CO_2$ + PCC + $CO_2$ compression emissions (kg $CO_2$/MW) | 27.84 | 25.29 |

Having additional steam for the process is very important to maintain the safety of the steam demanded by the process. Considering that the process is a refinery, the CHP plant provides a certain percentage of the steam to the process, while the remainder is provided by a conventional steam generator. The incorporation of the PTC would allow for a reduction in steam production and $CO_2$ emissions by the steam generator.

If increasing the electricity is more important than the additional saturated steam, then case 2 is an alternative because of the steam turbine.

Without the PTC, as shown in Figure 16, the total steam generated from the CHP plant was 1252.8 tonne/h of steam at 350 °C and 20 bar, of which 700 tonne/h was for the process, 119.1 tonne/h for the steam turbine to generate 23.22 MW, and 433.6 tonne/h was sent to the capture plant, first passing through a valve to reduce its pressure to 3.5 bar. The 433.6 tonne/h was attempered with 67.2 tonne/h of water in order to decrease the temperature.

When the PTC was incorporated in case 2, as presented in Figure 17, instead of sending 433.6 tonne/h to PCC from the CHP plant, this was reduced to 282.9 tonne/h. Thus, the steam to the steam turbine rose from 119.1 tonne/h to 269.8 tonne/h, the power generated in the steam turbine increased from 23.22 MW to 52.6 MW, and the net efficiency of the CHP plant with CCS increased from 65.4% to 68.21%.

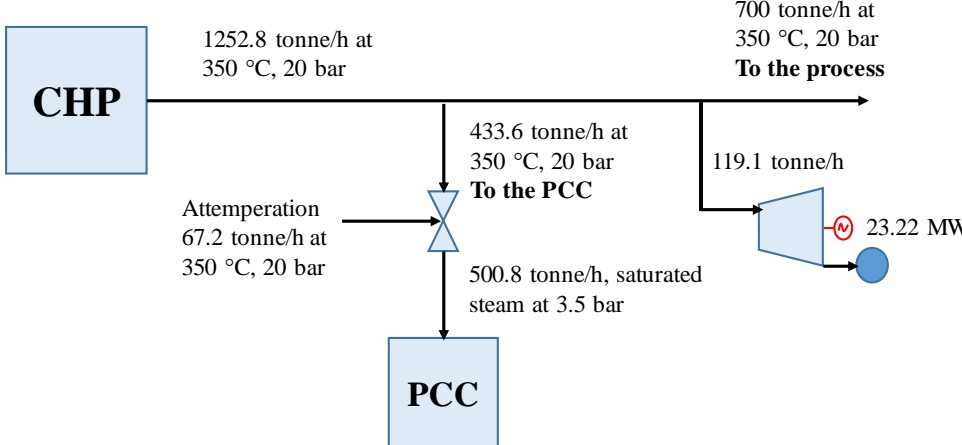

**Figure 16.** Steam mass balance of the CHP plant with $CO_2$ capture of case 2.

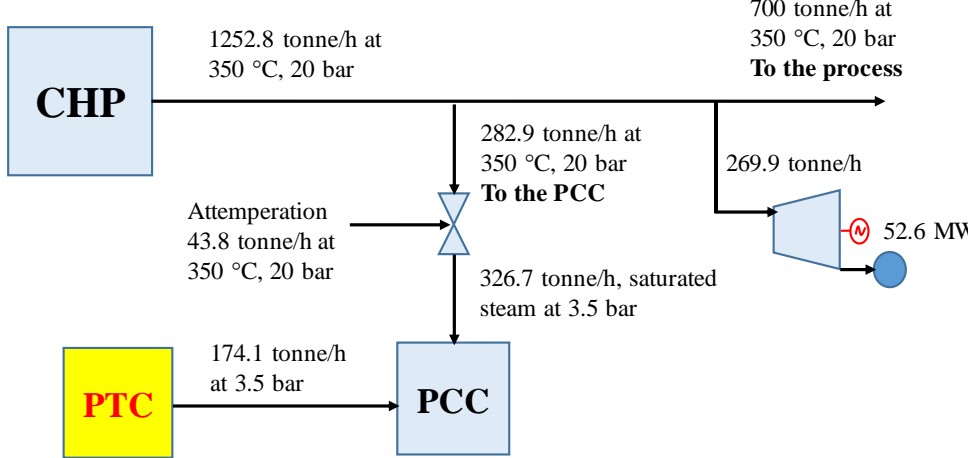

**Figure 17.** Steam mass balance of the CHP plant with $CO_2$ capture and PTC of case 2.

## 5. Conclusions

Three configurations of CHP with $CO_2$ capture were evaluated to identified the case with the highest efficiency. The CHP plant provided 700 tonne/h of steam at 20 bar and 350 °C to a petrochemical plant, leaving power generation free.

Case 3, which consists of two trains, each train has the configuration of one gas turbine with a heat recovery steam generator, presented the highest efficiency of 72.86% and generated 511.8 MW.

A PTC was incorporated in order to bring partially saturated steam at 3.5 bar to regenerate the MEA solvent. As a result, the efficiency of the CHP cycle increased from 72.86% to 80.18%. In addition, this additional saturated steam increased the flexibility of the CHP.

Although case 2 presents lower efficiency than case 3, because of the steam turbine, it brings the possibility to increase the amount of electricity instead of steam production. With the incorporation of the PTC, the power of the steam turbine increased from 23.22 MW to 52.6 MW, and the net efficiency from 65.4% to 68.21%.

The combination of CHP with $CO_2$ capture and solar thermal energy has a high potential for being an alternative for reducing $CO_2$ emissions in Mexico. From 2025 to 2036, the development perspectives for the CHP and the photovoltaic technology is expected to be around 9161 MW and 1043 MW, respectively; which would be incorporated into the Mexican electricity market. In addition, Mexico has high solar radiation around 5.7 kWh/m$^2$day.

The economic assessment is of great importance and will be investigated in future work.

**Author Contributions:** Conceptualization, A.M.A.C., O.A.J.S., N.V.L., M.R.P., J.O.A.A. and A.G.D.; methodology, A.G.D. and A.M.A.C.; software, A.M.A.C.; validation, A.G.D., A.M.A.C. and M.O.G.D.; formal analysis, A.M.A.C., O.A.J.S., N.V.L., M.R.P., J.O.A.A. and A.G.D.; investigation, A.M.A.C.; resources, A.G.D., A.M.A.C. and M.O.G.D.; writing—original draft preparation, A.M.A.C. and A.G.D.; writing—review and editing, A.M.A.C., A.G.D. and M.O.G.D.; supervision, O.A.J.S., N.V.L., M.R.P., J.O.A.A. and A.G.D. All authors have read and agreed to the published version of the manuscript.

**Funding:** This research received no external funding.

**Institutional Review Board Statement:** Not applicable.

**Informed Consent Statement:** Not applicable.

**Data Availability Statement:** No new data were created or analyzed in this study. Data sharing is not applicable to this article.

**Conflicts of Interest:** The authors declare no conflict of interest.

**Nomenclature**

| | |
|---|---|
| CHP | combined heat and power plant |
| CCS | carbon capture and storage |
| EOR | enhanced oil recovery |
| CSE | concentrate solar energy |
| CFE | Federal Commission of Electricity |
| DNI | direct normal irradiance |
| GT | gas turbine |
| HTF | heat transfer fluid |
| HRSG | heat recovery steam generator |
| IP | intermedia pressure |
| LCS | levelised cost of steam |
| LHV | low heat value |
| LP | low pressure |
| MEA | monoethanolamine |
| MW | megawatts |
| NGCC | natural gas combined cycle power plant |
| PTC | parabolic-trough collector |
| PTCSP | parabolic-trough collector solar plant |
| PTCSTPP | parabolic-trough-concentrating solar thermal power plant |
| PCC | post-combustion carbon capture |
| Qt | total thermal heat |
| SDC | Stirling dish collector |
| SPT | solar power tower |
| SAM | system advisor model |
| TES | thermal energy storage |

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
