# Peer review of "Optimisation of an Integrated System: Combined Heat and Power Plant with CO2 Capture and Solar Thermal Energy"

_processes, doi:10.3390/pr11010155_

Round 1
Reviewer 1 Report
The authors compared the three configurations of a combined heat and power with post-combustion CO2 capture. This review gives a systematic description on this important issue. Therefore, I recommend publishing this work after a minor revision.
1. In the conclusion, the authors should supplement the development perspectives in the combination of heat and power plant with CO2 capture and solar thermal energy.
2. Please improve the quality of some Figures.
3. Authors need to revise and proofread the manuscript to improve the English language.
Reviewer 2 Report
The authors evaluated the various design configurations of a combined heat and power (CHP) with post-combustion CO2 capture. The manuscript subject is good however, in order to improve the quality and readers' clarity following suggestions are recommended.
1) Introduction section is good and reference added are also good.
2) In section 2.1 where combined heat and power plant description is mentioned - various configuration of CHP is mentioned; typically GT selection, HRSG selection needs proper justification. How and why particular GT and HRSG are selected? add justification
3) in section 2.3: the three case studies mentioned are simulated using Aspen Plus, why this software is used. add justification. Also add details of software like company supplied, version, etc
4) Figure 6, 7 and 8 - add legends for the curves plotted
5) Figure 13 need clarity, improve it for readers readibility
6) Recommanded to add annex for system analysis sample calculations. it will help to understand the calculations made
7) Is any experimental or practical results available?
8) conclusion can be improved further by adding future scope.
Round 2
Reviewer 2 Report
all questions are neatly addresed.